# MHA-ConvLSTM Dam Deformation Prediction Model Considering Environmental Volume Lag Effect

Hepeng Liu [1], Denghua Li [2,3,*] and Yong Ding [1,*]

1 School of Science, Nanjing University of Science and Technology, Nanjing 210094, China; liu130896@njust.edu.cn
2 Nanjing Hydraulic Research Institute, Nanjing 210029, China
3 Key Laboratory of Reservoir Dam Safety, Ministry of Water Resources, Nanjing 210029, China
* Correspondence: dhli@nhri.cn (D.L.); njustding@163.com (Y.D.)

**Abstract:** The construction of a reasonable and reliable deformation prediction model is of great practical significance for dam safety assessment and risk decision-making. Traditional dam deformation prediction models are susceptible to interference from redundant features, weak generalization ability, and a lack of model interpretation. Based on this, a deformation prediction model that considers the lag effect of environmental quantities is proposed. The model first constructs a new deformation lag influence factor based on the plain HST model through the lag quantization algorithm. Secondly, the attention and memory capacity of the model is improved by introducing a multi-head attention mechanism to the features of the long-time domain deformation influence factor, and finally, the extracted dynamic features are transferred to the ConvLSTM model for learning, training, and prediction. The results of the simulation tests based on the measured deformation data of an active dam show that the introduction of the deformation lag factor not only improves the interpretation of the prediction model for deformation but also makes the prediction of deformation more accurate, and it can improve the evaluation indexes such as RMSE by 50%, the nMAPE by 40%, and $R^2$ by 10% compared with the traditional prediction model. The combined prediction model is more capable of mining the hidden features of the data and has a deeper picture of the overall peak and local extremes of the deformation data, which provides a new way of thinking for the dam deformation prediction model.

**Keywords:** hysteresis; ConvLSTM; attention mechanism; prediction model; dam deformation





## 1. Introduction

In recent years, a global slowdown in water and hydropower construction has led countries to pay attention to the management of existing water and hydropower facilities [1], so the effective development of water "four pres" technology has become a key engineering problem in the field of dam safety monitoring [2–4].

Dams are subjected to a variety of factors such as the nature of the dam foundation engineering, temperature conditions, structural design, and their own loading during their complex and variable service, and their properties will gradually deteriorate with increasing service life [5–7]. This deterioration may lead to an increase in the probability of accidents in dams, which in turn may cause serious property damage and casualties in the surrounding towns [8–10]. Therefore, the focus of research has shifted to how to reasonably and effectively construct dam deformation early warning and forecasting models. Among them, it is of great practical importance to rely on actual dam measurement data to study the nonlinear relationship between dam deformation and each deformation characteristic factor, to establish a reliable dam deformation monitoring model, to accurately predict the deformation of dams, and to comprehensively understand their operational status to ensure the long-term safety of dam projects [11].

As the analysis of deformation monitoring information has developed toward a more in-depth field, many scholars have proposed some new methods and techniques to monitor the deformation of dams. Among them, optimization combination models, digital filtering, and principal component regression are widely used in the field of dam safety monitoring [12–14]. At present, in practical engineering, deformation monitoring models can be divided into three main categories, including statistical models, deterministic models, and machine learning models [15,16].

Statistical models are based on statistical analysis and the modeling of large amounts of deformation monitoring data to derive probability distributions and trend predictions of the deformation behavior of dams [17]. Typical statistical models are the hydrostatic-thermal-time (HTT) model [18] and the hydrostatic-season-time (HST) model [19]. Among them, the HTT model is applicable when the temperature measurement points of the dam body and foundation can fully describe the variation in its temperature field, and the measured values of each temperature measurement point are used as the temperature components [18]. The HST model, on the other hand, uses harmonic factors to describe the temperature components when the temperature monitoring data inside the dam are lacking or insufficient, and they usually coincide with the deformation of the dam in most cases [19].

Deterministic models are physical models based on the structural properties, material properties, and loads of a dam. By modeling and analyzing the mechanical behavior of a dam, its deformation response and safety conditions can be predicted [20]. Deterministic models usually use numerical computational methods, such as finite element analysis [21]. However, traditional deterministic modeling methods have several limitations and challenges. First, the implementation of the method may require significant time and effort because of the need to build complex models and perform detailed numerical calculations. Second, some assumptions and simplifications are often required for the geometry and boundary conditions of the dam, which may negatively affect the accuracy of the predictions [22,23]. In addition, the lack of long-term monitoring data makes it difficult for these models to accurately consider the long-term deformation behavior of dams [24].

With the continuous progress of machine learning technology, its application in dam deformation prediction and analysis has become a hot spot for research. Compared with traditional mathematical models, machine learning models can better handle complex nonlinear relationships and learn the patterns and laws between deformation and various influencing factors from a large amount of measured data [25,26].

Stojanovic et al. [27] developed a neural-network-based dam deformation modeling system that can adapt to the dynamic changes in the dam observation data by updating the algorithm parameters in real time and effectively improving the adaptive nature of the model. This can reduce the influence of artificial parameters on model accuracy while ensuring the timeliness of parameter selection. Zhu et al. [28] used adaptive theory to optimize the artificial bee colony algorithm and combined it with a back propagation (BP) neural network model to accurately predict the deformation sequences of high arch dams and provide a safety state analysis of dam structures with Gourine's [29] combined singular spectrum analysis and ANNs to study the influencing factors of dam displacement, and they achieved good results. Wei et al. [30] considered the influence that complex nonlinearities in the residual sequence would have on the prediction accuracy in the modeling process of deformation prediction, and they proposed a combined prediction model, which integrated wavelet decomposition, neural networks, and integrated moving average autonomy on the basis of traditional statistical models. Barzaghi [31] used a GNSS to analyze the deformation of the Eleonora DArborea dam and demonstrated that GNSS technology could reflect the displacement of the dam in both time and space. For the highly nonlinear problem of temperature factor, based on the deformation prediction model constructed using radial basis neural network technology, the kernel principal component analysis method was introduced to reduce the dimensionality of the temperature effect quantity, and the effectiveness of the model was verified in engineering applications.

However, neural network methods are prone to fall into local optimal solutions in the prediction process due to the gradient-descent-based training [32,33], and their generalization ability still needs to be improved. For this reason, Kang et al. [34], Li et al. [35], and Fan et al. [36] applied the extreme learning machine (ELM) model to the dam deformation prediction. The ELM model uses the least squares method for the model solution, which avoids the impact of iterative computation on network accuracy. The effectiveness and superiority of the extreme learning machine model were demonstrated by comparing it with statistical models and conventional artificial neural network methods and by validating it in engineering examples [37,38].

In summary, a combined MHA-ConvLSTM (Multi-Head Attention-ConvLSTM) model that takes the environmental volume lag into account is suggested. By including lag variables to strengthen the explanation of the prediction model and by merging the attention mechanism and convolution technique, they seek to boost the performance of the prediction model in memorization, depiction, and mining features. Finally, by comparing and assessing the prediction effect with other models or benchmark models for actual engineering data, the viability of introducing the lag component and the efficacy of the combined prediction model are confirmed.

The remainder of this essay is structured as follows. The deformation hysteresis quantification algorithm's fundamental principles and implementation procedure are covered in Section 2. The proposed hybrid prediction model for dam deformation, MHA-ConvLSTM, is also provided along with its creation method. Section 3 gives a full examination of the findings from each of the three validation procedures for the model, including the feasibility validation of the hysteresis factor addition, the ablation validation, and the validity and generalization validation. Section 4 concludes the entire essay and presents a plan for more research.

## 2. Methodology

### 2.1. Deformation Hysteresis Quantification Algorithm

In order to quantify the deformation lag time, which is primarily influenced by environmental quantities, this paper proposes a deformation lag quantification algorithm by analyzing the fluctuations and phase differences between the environmental quantities (water level, temperature, etc.) and the measured data of the effect quantity (dam deformation). This approach only considers the water level ($H$) and dam deformation ($\delta$), as this paper employs HST (hydrostatic-season-time) as the basis model. Due to the water level variation factor, the dam deformation data and water level data are similar, but there is a slight phase difference between the two.

Therefore, the fundamental concept behind the lag quantification approach is to first employ data decomposition and reconstruction techniques to minimize the negative effects of irregular terms on the phase computation and then to compute the phase cross-entropy of both to determine the lag time. These are the precise stages in this algorithm:

Step 1: Data pre-processing. The monitoring data were normalized to reduce the order-of-magnitude differences between different physical quantities, and the normalization equation was expressed as Equation (1). To reduce the problem of the inability to perform cross-entropy calculations due to low data density, the hot card filling method [39] was used for the monitoring data to make full use of the contextual linkage of the monitoring data for the regression interpolation of missing values, so that the monitoring data have a first-order continuous type.

$$X_i^{norm} = \frac{X_i - X_{\min}}{X_{\max} - X_{\min}} \tag{1}$$

In the above equation, $X_i^{norm}$ is the normalized data, $X_i$ is the pre-normalized data, and $X_{\max}$ and $X_{\min}$ are the maximum and minimum values of the monitored data.

Step 2: Breaking down and rebuilding monitoring data. VMD decomposition was used to decompose the monitoring data, and the high-frequency information obtained from the decomposition was retained and fused, reducing the detrimental effects of low-frequency

information on the subsequent calculation of phase cross-entropy. The specific steps of the VMD decomposition reconstruction technique are described in the literature. [40].

Step 3: Hysteresis quantization calculation. For the reconstructed monitoring information, firstly, one set of data was transformed to the frequency of another set of data by using discrete Fourier transform (DFT) [41–45]. Next, the lag was calculated using the phase cross-entropy calculation method. The formula for calculating the phase cross-entropy *CE* is shown in Equation (2)

$$CE = -\sum[P(i,j) \times \log(P(i,j))] \tag{2}$$

where $P(i,j)$ is the probability of simultaneous occurrence of the time series $i$ and $j$ with a phase difference of $j$. *CE* denotes the phase cross-entropy between the time series $i$ and $j$. The phase difference between the time series $i$ and $j$ can be obtained by calculating the phase cross-entropy of the time series. If the phase difference is zero, they are synchronized. If the phase difference is positive, it means that the time series $i$ is lagging behind the time series $j$. If the phase difference is negative, then the time series is lagging behind the time series $ji$.

### 2.2. MHA-ConvLSTM Combined Prediction Model
2.2.1. Multi-Head Attention Mechanism (Multi-Head Attention)

Traditional recurrent neural networks (RNNs) struggle to capture long-range dependencies in sequences when working with extended time domain temporal data. The model's performance can be enhanced by using the attention technique to assist the model in more accurately capturing these dependencies. The fundamental goal of the attention mechanism is to direct attention to the information that is most important for the current task among a great amount of incoming data while reducing or even removing attention to other aspects. By addressing the issue of information overload, this strategy increases the task processing's effectiveness and precision. The operations for the typical attention mechanism module are query ($Q$), key ($K$), and value ($V$). In order to get the overall weights and outputs, the attention weights must first be determined using $Q$ and $K$, and then they must be applied to $V$. According to Equation (3), the output vector is specifically calculated for the input matrices $Q$, $K$, and $V$.

$$Attention(Q,K,V) = Soft\max\left(\frac{QK^T}{\sqrt{d_k}}\right)V \tag{3}$$

Equation (3) in $Q \in R^{n \times d_k}, K \in R^{m \times d_k}, V \in R^{m \times dv}$.

The multi-head attention mechanism is optimized and improved from multiple parallel attention mechanisms, which not only improves the computational speed but also enables the network to adaptively select important data features to train the network model, increasing the diversity of extracted features; the collaboration between multiple heads helps the network to learn deeper data features, while the collaboration between multiple heads also improves the accuracy of the model prediction. The mathematical expression of the multi-head attention mechanism as Equation (4) is:

$$MultiHead(Q,K,V) = Concat(head_1, \cdots, head_k, \cdots, head_n)W^O \tag{4}$$

In Equation (4), $Concat(\cdot)$ denotes the vertical stitching operation of the matrix and $W^O$ denotes the weight matrix $W^O \in R^{d_v \times d_m}$.

2.2.2. Convolutional Long- and Short-Term Memory Neural Network (ConvLSTM)

LSTM and CNN are two commonly used neural network models; each of them is good at extracting different types of features. LSTM is mainly used to extract features between adjacent sequence data, so it is effective in dealing with time series prediction problems. CNN, on the other hand, is able to extract effective bias features as feature vectors and

obtain more structural information. In order to fully utilize the advantages of both models, a new model, ConvLSTM, is proposed.

ConvLSTM combines the advantages of LSTM and CNN and is able to extract spatio-temporal features simultaneously instead of a single temporal feature. Unlike the traditional single-step spatio-temporal prediction, ConvLSTM uses a multi-step spatio-temporal prediction and can predict future data trends more accurately. Therefore, this paper adopts the ConvLSTM model to mine the hidden local features and spatio-temporal features of the data. The specific formula of ConvLSTM is shown in the following Equations (5)–(9):

$$Z^{t,l} = \sigma\left(\widetilde{W}_{XZ}^{l} * X^{t,l} + \widetilde{W}_{HZ}^{l} * H^{t-1,l} + \widetilde{W}_{CZ}^{l} \circ C^{t-1,l} + \widetilde{b}_{Z}^{l}\right) \tag{5}$$

$$r^{t,l} = \sigma\left(\widetilde{W}_{XR}^{l} * X^{t,l} + \widetilde{W}_{HR}^{l} * H^{t-1,l} + \widetilde{W}_{CR}^{l} \circ C^{t-1,l} + \widetilde{b}_{R}^{l}\right) \tag{6}$$

$$C^{t,l} = z^{t,l} \circ \tanh\left(\widetilde{W}_{XC}^{l} * X^{t,l} + \widetilde{W}_{HC}^{l} * H^{t-1,l} + \widetilde{b}_{C}^{l}\right) + r^{t,l} \circ C^{t-1,l} \tag{7}$$

$$o^{t,l} = \sigma\left(\widetilde{W}_{XO}^{l} * X^{t,l} + \widetilde{W}_{HO}^{l} * H^{t-1,l} + \widetilde{W}_{CO}^{l} \circ C^{t,l} + \widetilde{b}_{O}^{l}\right) \tag{8}$$

$$H^{t,l} = o^{t,l} \circ \tanh\left(C^{t,l}\right) \tag{9}$$

where $*$ denotes the convolution operator, $\circ$ denotes the Hadmard function, and $\sigma$ denotes the Sigmoid function, where $\widetilde{W}_{XZ}^{l}$, $\widetilde{W}_{HZ}^{l}$, $\widetilde{W}_{CZ}^{l}$, $\widetilde{W}_{XR}^{l}$, $\widetilde{W}_{HR}^{l}$, $\widetilde{W}_{CR}^{l}$, $\widetilde{W}_{XC}^{l}$, $\widetilde{W}_{HC}^{l}$, $\widetilde{W}_{XO}^{l}$, $\widetilde{W}_{HO}^{l}$, and $\widetilde{W}_{CO}^{l}$ are the bias parameters of the convolutional long- and short-term neural network in the $l$ layer.

### 2.2.3. Combined Prediction Model

The combined prediction model based on MHA-ConvLSTM is intended to solve the problems of poor feature extraction ability, insufficient local detail portrayal ability, and low prediction accuracy of traditional prediction models for longer time domain data.

The specific steps are as follows:

Step 1: Input the water level data and deformation data into the hysteresis quantization algorithm to calculate the hysteresis.

Step 2: Generate a new impact factor dataset by constructing lagging factors based on the quantification results.

Step 3: Feature extraction of the input impact factor dataset via the multi-headed attention mechanism.

Step 4: The extracted data features are trained and predicted by using ConvLSTM.

Step 5: The evaluation of predicted effects.

The specific flow of the hybrid prediction model based on MHA-ConvLSTM is shown in Figure 1.

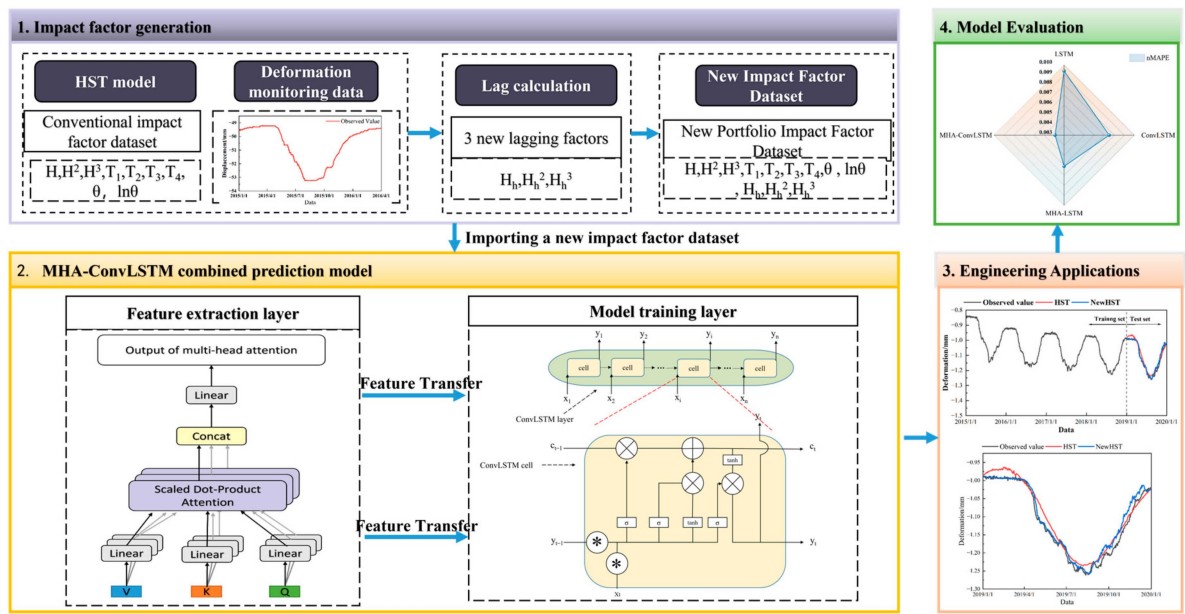

**Figure 1.** Flow chart of MHA-ConvLSTM model.

### 2.3. Predictive Performance Quantifiers

This study suggests choosing RMSE (root mean square error), nMAPE (normal-ized mean absolute percentage error), and $R^2$ (coefficient of determination) as quantitative indicators of prediction performance in order to assess the prediction performance of the prediction algorithm at various levels. In Equations (10)–(12), the precise formulas for the aforementioned metrics are displayed.

$$RMSE = \sqrt{\frac{1}{n}\sum_{i=1}^{n}\left(y_m^i - y_p^i\right)^2} \tag{10}$$

$$nMAPE = \frac{1}{n}\sum_{i=1}^{n}\left|\left(\frac{y_m^i - y_p^i}{|y_m^i|_{\max}}\right)\right| \tag{11}$$

$$R^2 = 1 - \frac{\sum\limits_{i=1}^{n}\left(y_m^i - y_p^i\right)^2}{\sum\limits_{i=1}^{n}\left(y_m^i - \bar{y}_m\right)^2} \tag{12}$$

In the above equation, $n$ is the total number of data samples, $y_m^i$ and $y_p^i$ are the measured and predicted values of $i$, and $\bar{y_m}$ is the average of $y_m$. In these metrics, RMSE is the square root of the mean of the squared difference between the measured and predicted values. The smaller the RMSE, the closer the prediction result is to the actual value and the more accurate the prediction algorithm is. $nMAPE$, as an improvement of $MAPE$, is guaranteed that the denominator is not zero. For $nMAPE$, the smaller it is, the better the prediction performance is represented. Additionally, $R^2$ indicates how well the predicted value fits the actual value. The closer $R^2$ is to 1, the better the predicted value fits the actual value.

## 3. Case Studies

### 3.1. Experimental Data Sources

The real measurement dataset of an operating dam is chosen as the case study basis for an in-depth investigation and model analysis in order to confirm the viability and generalizability of the suggested method. The project is situated in the Chinese region of

Xinjiang Uygur Autonomous Region. The project's principal development task is power generation while taking into account the needs for downstream flood control. The total storage capacity is 125 million m$^3$, the regulating storage capacity is 72.4 million m$^3$, the normal storage level is 1649 m, and the maximum dam height is 110 m. An automated safety inspection system, consisting of several instruments measuring temperature, water level, seepage pressure, deforestation, etc., is deployed on the surface and inside the dam to guarantee that it operates normally during its service life. The system, which consists of a variety of sensors for measuring temperature, water level, seepage pressure, deformation, and other variables, realizes the integrated function of automatic monitoring, computation, transmission, and storage, which enables data-driven forecasting and the early warning modeling of dam property changes.

As the test case for this study, the EX1–2, EX1–3, EX2–3, EX2–4, and water level data within the elevation range of 1451.52–1470 m in the left 0 + 045 dam section of this dam were chosen. The monitoring data were obtained from the period of 1 January 2015 to 1 January 2020. According to the actual engineering requirements, the data collection frequency of these monitoring instruments is once per day. The horizontal displacement monitoring device employed in this dam is called EX. Figure 2 displays the location details of each measurement point, Figure 3 displays the time course curve of each measurement site's deformation data, and Figure 4 displays the time course curve of the water level data.

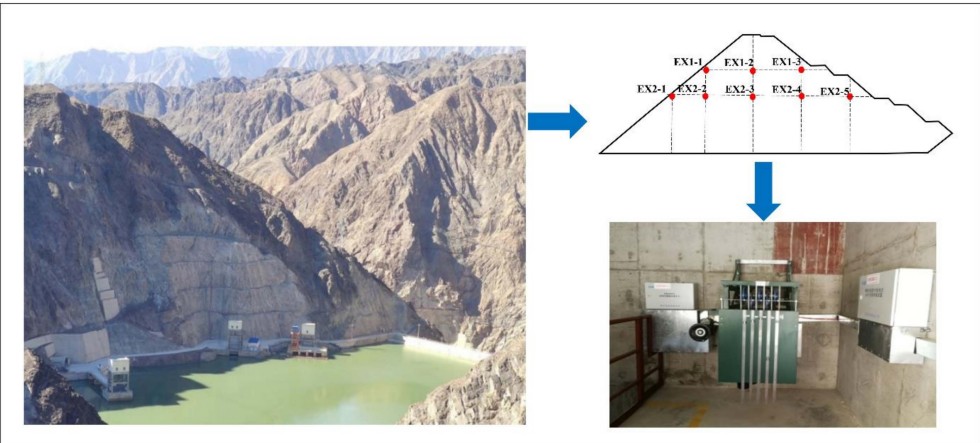

**Figure 2.** Basic information about the study dam.

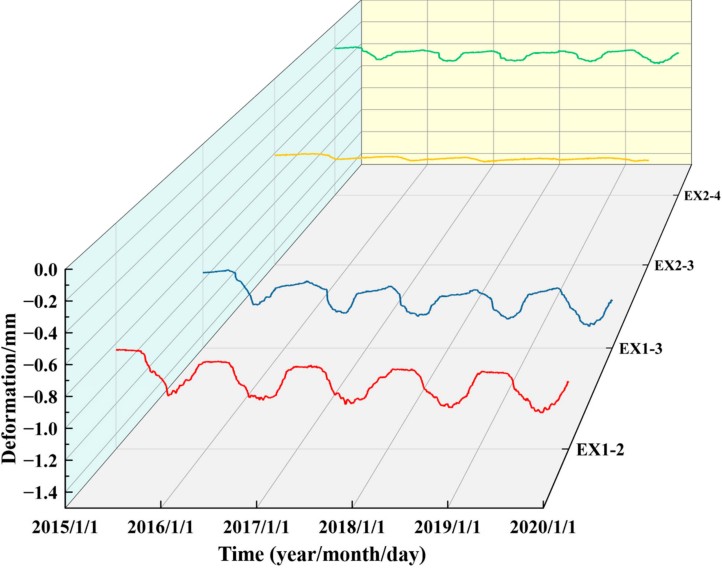

**Figure 3.** Time course curve of deformation data.

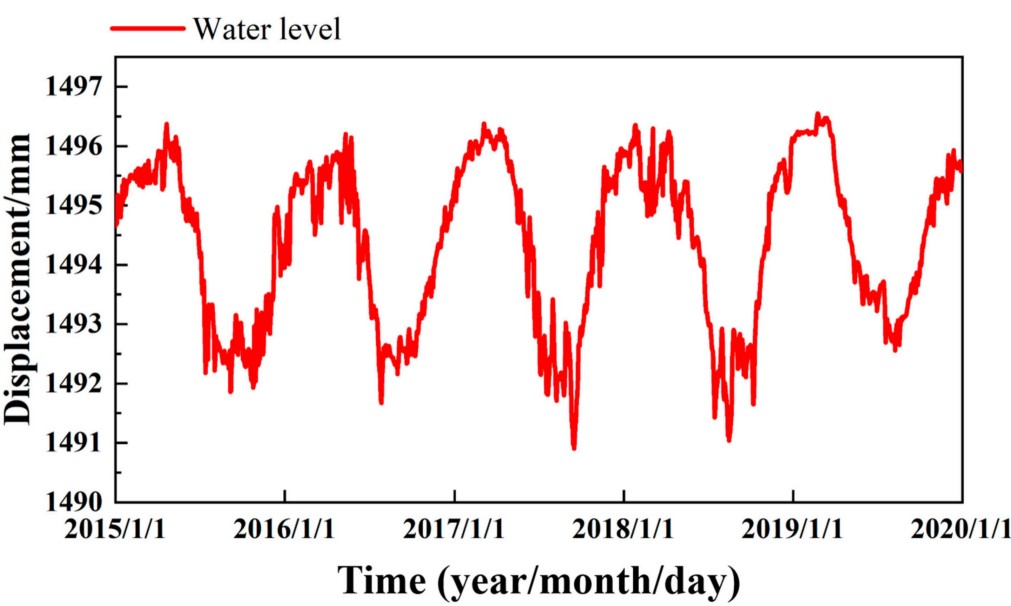

**Figure 4.** Time course curve of water level data.

### 3.2. Feasibility Validation of Hysteresis Factor Addition

The hydrostatic-season-time (HST) model, a classical and well-accepted mathematical model for dam deformation prediction, predicts the deformation of dams by considering the hydrostatic pressure, seasonal factors, and time dependence of the dam. The HST model provides an alternative method to infer the dam temperature variation and correlate it with dam deformation behavior when temperature monitoring information in the dam is not sufficient. The key to the HST model is the application of the harmonic factor. The harmonic factor is a mathematical technique used to characterize the periodicity of temperature variables. By performing spectral analysis and harmonic decomposition on temperature data, different harmonic components that reflect the seasonal and periodic patterns of temperature variation can be identified. By correlating these harmonic factors with the dam deformation data, a quantitative relationship between the dam deformation and temperature variation can be established. In most cases, the predictions of the HST model coincide with the dam deformation.

According to the HST model described, the deformation of any point of the dam ($\delta$) can be decomposed into a water pressure component ($\delta_H$), a temperature component ($\delta_T$), and an aging component ($\delta_\theta$). The specific equation is shown in Equation (13)

$$\delta = \delta_H + \delta_T + \delta_\theta = a_0 + \sum_{i=1}^{n} a_i H_i + \sum_{i=1}^{m} \left( b_{1i} \sin \frac{2\Pi i t}{365} + b_{2i} \cos \frac{2\Pi i t}{365} \right) + c_1 \theta + c_2 \ln \theta \quad (13)$$

where $a_0$ is the constant term, $a_i$ is the regression coefficient of the water pressure component, $b_{1i}$ and $b_{2i}$ are the regression coefficients of the temperature component, $c_1$ and $c_2$ are the regression coefficients of the time-dependent component, $n$ is the regression coefficient of the gravity dam ($n$ is 3 depending on the dam type), $H$ is the reservoir level, $m$ is 2 or 3 depending on the temperature at the project site (this paper takes 2), and $\theta$ is the cumulative number of days from the day of observation to the starting date divided by 100. In summary, the influence factor of the initial HST model in this paper is $H, H^2, H^3, \sin \frac{2\Pi t}{365}, \sin \frac{4\Pi t}{365}, \cos \frac{2\Pi t}{365}, \cos \frac{4\Pi t}{365}, \theta, \ln \theta$.

To verify the effect of adding the lag factor, the lag quantification in Section 2.1 was performed with EX1-2 as an example. The case of EX1-2 with water level decomposed via VMD is shown in Figure 5. The reconstructed case is shown in Figure 6.

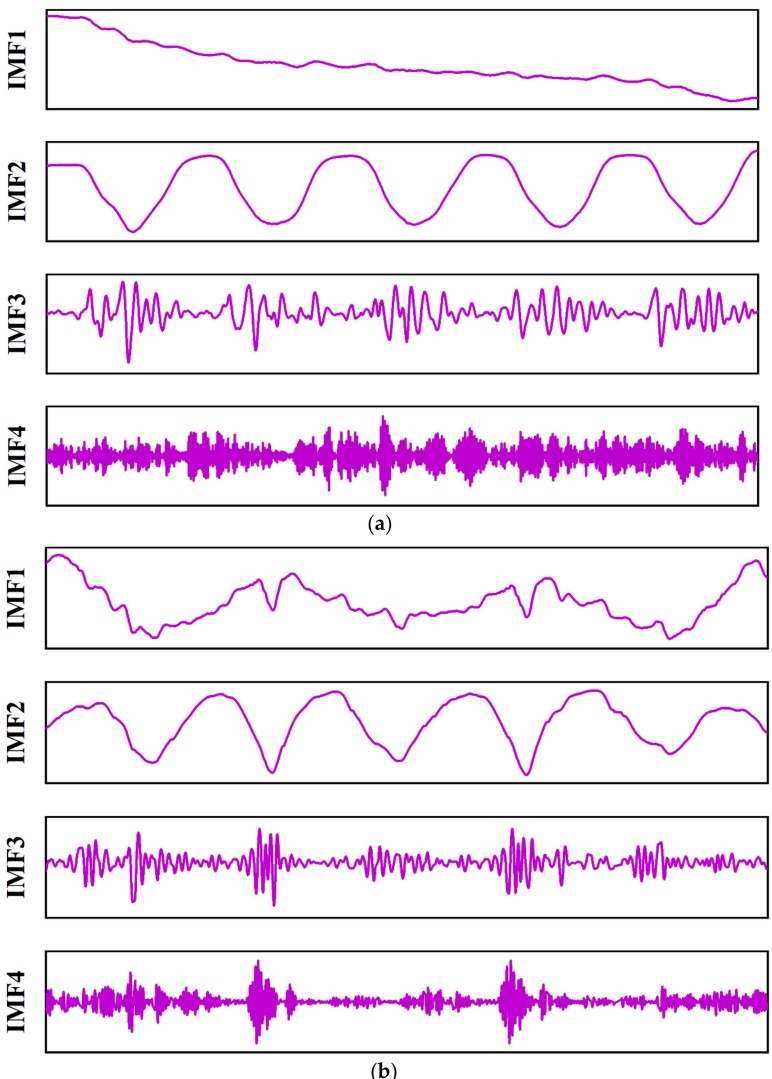

(a)

(b)

**Figure 5.** (**a**) Information of EX1-2 deformation data decomposition. (**b**) Water level data decomposition information.

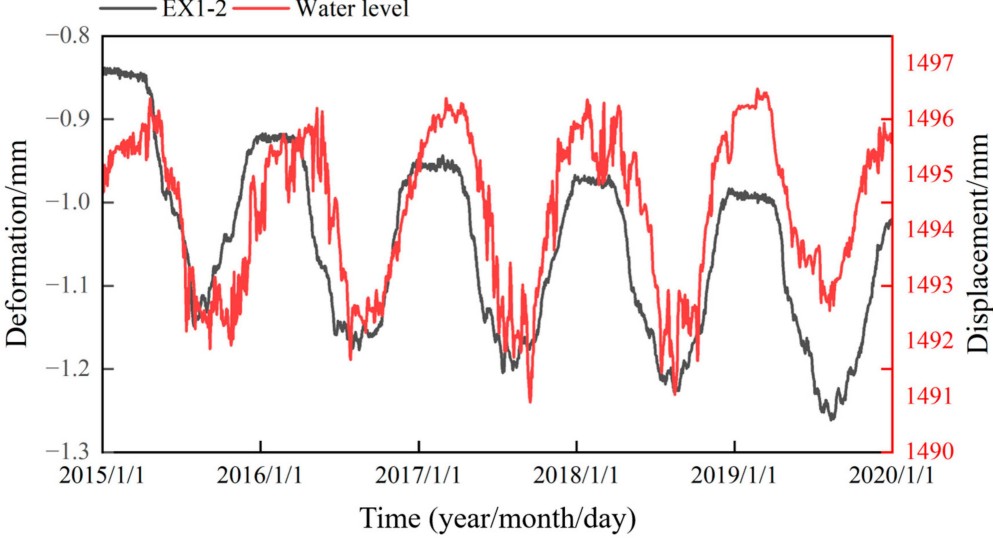

**Figure 6.** Time course curves of reconstructed deformation and water level data.

The lag between the EX1–2 and the water level after reconstructing the data according to Equation (2) in Section 2.1 can be calculated, and EX1–2 deformation lags 11 days behind the water level. Therefore, a new lag factor is introduced with reference to Equation (13), i.e., $H_{11}, H_{11}^2, H_{11}^3$, to form a combination of impact factors for the HST model considering lag, i.e., $H, H^2, H^3, \sin\frac{2\Pi t}{365}, \sin\frac{4\Pi t}{365}, \cos\frac{2\Pi t}{365}, \cos\frac{4\Pi t}{365}, \theta, \ln\theta, H_{11}, H_{11}^2, H_{11}^3$.

To verify the feasibility of the addition of the lag factor, the plain LSTM model was used to predict the HST model for the above two combinations. Following the principle of selecting 80% of the data as the training set and 20% as the validation set, the data from 1 January 2015 to 1 January 2019 were chosen as the training set, and the data from 1 January 2019 to 1 January 2020 were chosen as the test set. The prediction results are shown in Figure 7 below, the box line plot of the prediction residuals is shown in Figure 8, and the details of the evaluation indexes are shown in Table 1.

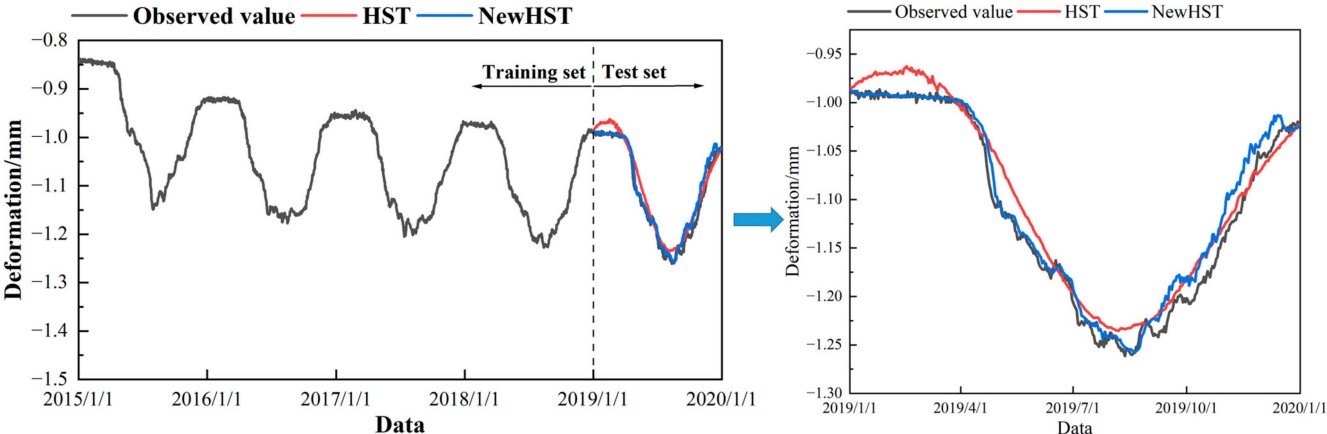

**Figure 7.** Prediction with and without lag factor model.

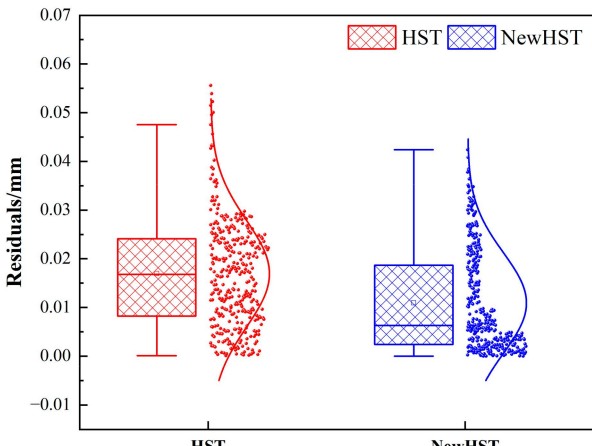

**Figure 8.** Distribution of predicted residuals.

**Table 1.** Comparison of prediction evaluation indicators with and without lag factor models.

| Models | nMAPE | RMSE | $R^2$ |
|---|---|---|---|
| HST-LSTM | 0.0101 | 0.0154 | 0.9741 |
| NewHST-LSTM | 0.0086 | 0.0121 | 0.9853 |

From the analysis of the above results, it is clear that the HST model with lag quantization can more perfectly capture the lag effect of environmental quantities for the EX1–2 time series compared with the HST model without lag quantization. The prediction curves

under the plain HST model, although smooth, are poorly fitted to the real data and do not portray the local fluctuations of the measured values, especially in the time period when the prediction starts (1 January 2019–1 April 2019). The contrast between the two is more obvious because the input variables of the plain HST model do not contain enough information to feed back the nonlinear characteristics of the deformation, thus leading to some apparent deviation from the measured data and a local distortion. Meanwhile, the box plot of residuals in Figure 8 shows that the median and mean of the residuals are smaller and the distribution of residuals is denser for the HST model with lags considered compared with the HST model without lag quantification, which further verifies the feasibility of introducing lag factors in the HST model.

### 3.3. Ablation Verification

Because the multi-headed attention mechanism and CovLSTM are both included in this hybrid model, we compared the prediction performance of the plain LSTM, plain LSTM + multi-headed attention mechanism, and plain LSTM + CovLSTM considering lags using the EX1–3 time series as an example to confirm the improvement performance of the hybrid model over the plain model.

Figure 9 below depicts the prediction effect, Figure 10 depicts the prediction residual box line plot, Figure 11 depicts the evaluation index effect plot, and Table 2 depicts the improvement level of each model over plain LSTM.

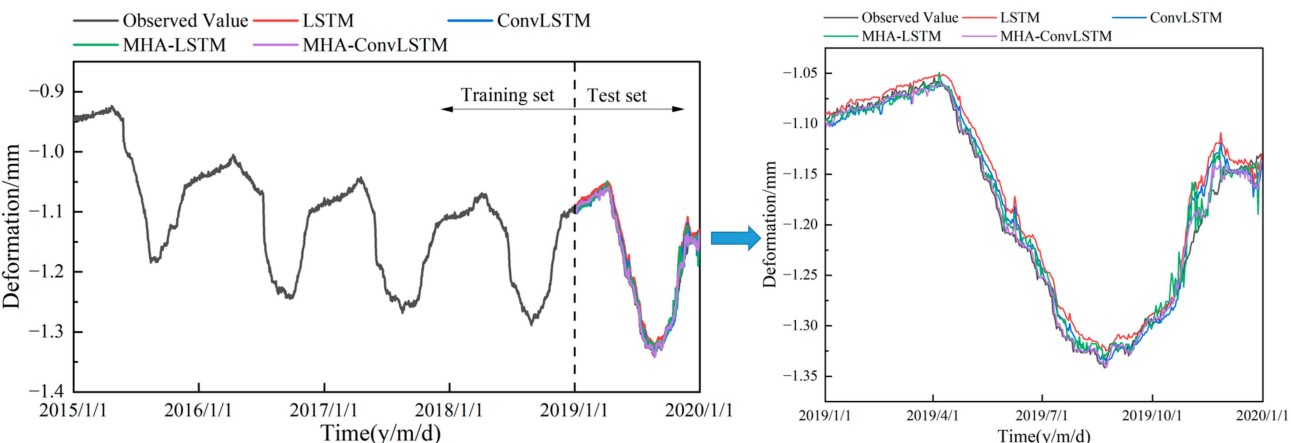

**Figure 9.** Prediction with and without lag factor model.

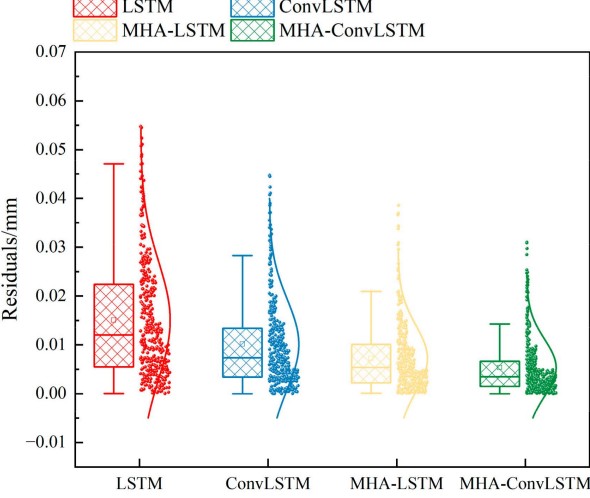

**Figure 10.** Distribution of predicted residuals.

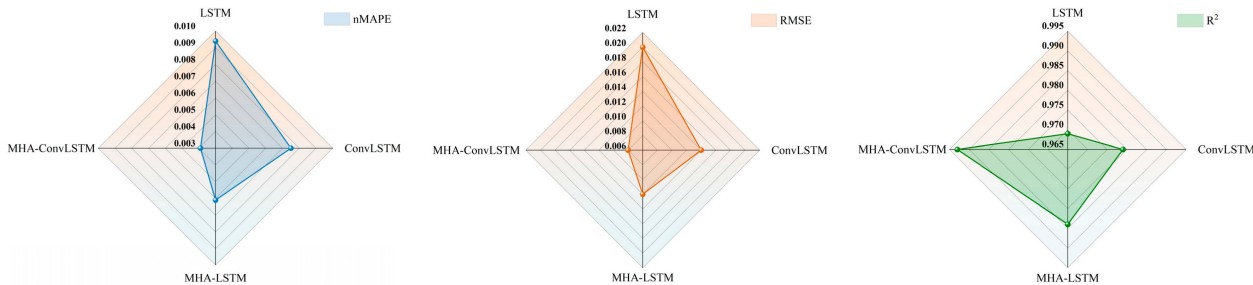

**Figure 11.** Comparison of evaluation indexes of each prediction model.

**Table 2.** The lifting degree of each model compared with plain LSTM.

| Elevation | nMAPE | RMSE | $R^2$ |
|---|---|---|---|
| ConvLSTM | 19.1% | 47.7% | 1.1% |
| LSTM + MHA | 35.7% | 58.4% | 1.6% |
| MHA-ConvLSTM | 57.5% | 72.5% | 2.5% |

The results demonstrate that the addition of the attention mechanism significantly enhances the ability of the LSTM model to remember and utilize features, which greatly improves the prediction accuracy of the model with at least a 50% improvement in RMSE. This is in contrast to the plain LSTM model, which is more sensitive to redundant information when facing long-time domain data. The findings also demonstrate that for neural network models, a more effective feature learning model can raise the model's predictive accuracy. The fit of the model to the periodic changes and lagging characteristics of the dam deformation data is improved, and the $R^2$ is improved by at least 1.5%, further proving the viability of combining the multi-headed attention mechanism with ConvLSTM. ConvLSTM is better at feature learning than LSTM and can more effectively use the important features fed with the attention mechanism.

### 3.4. Validation of Prediction Model Validity and Generalization

It is proposed to take EX2–3 and EX2–4 as examples to fully verify the effectiveness and generalization of this prediction model by comparing the prediction results with the current mainstream prediction models (BP, Ridge, XGBoost).

The EX2–3 prediction effect is shown in Figure 12a below, the prediction residual box line diagram is shown in Figure 12b, the evaluation index effect diagram is shown in Figure 13, and the improvement degree of each model compared with BP is shown in Table 3.

**Table 3.** Lifting degree of each model compared with BP.

| Elevation | nMAPE | RMSE | $R^2$ |
|---|---|---|---|
| Ridge | −1.3% | −0.7% | −0.2% |
| XGBoost | 62.3% | 58.3% | 9.4% |
| MHA-ConvLSTM | 69.6% | 66.4% | 10.1% |

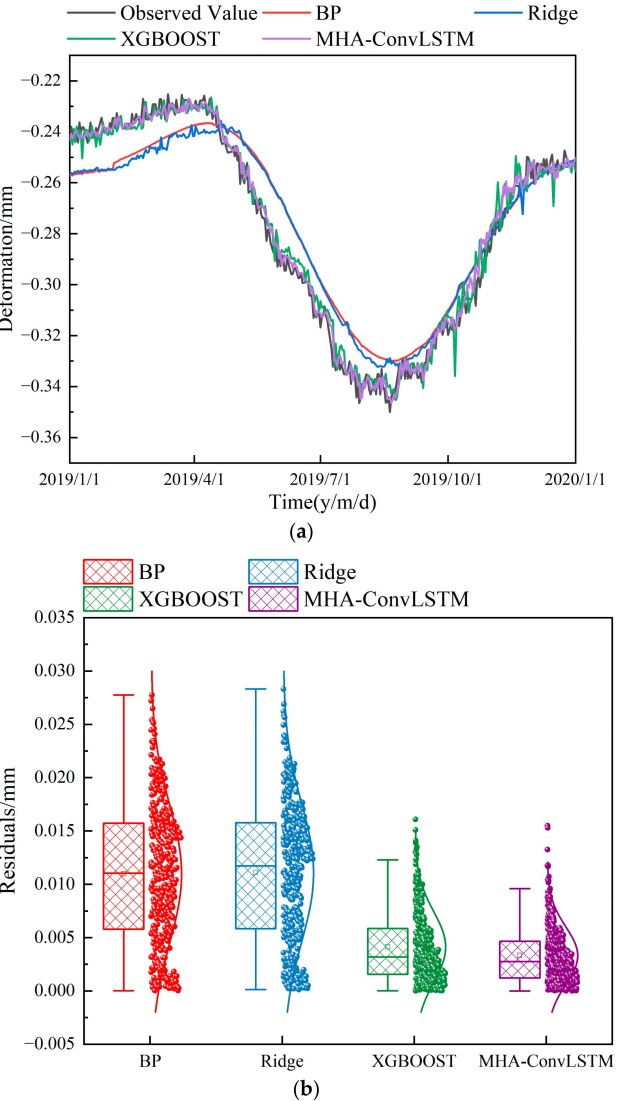

**Figure 12.** (**a**) Forecast of different forecasting models. (**b**) Distribution of residuals for each model prediction.

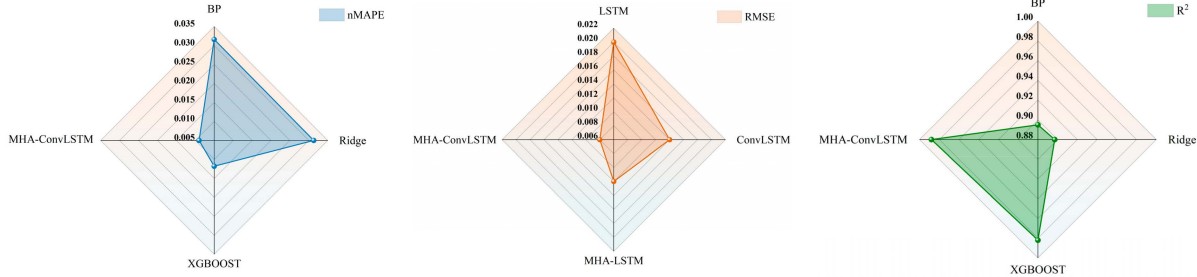

**Figure 13.** Comparison of evaluation indexes of each prediction model.

The effect of EX2–4 prediction is shown in Figure 14a below, the residual box line of prediction is shown in Figure 14b, and the effect of the evaluation index is shown in Figure 15.

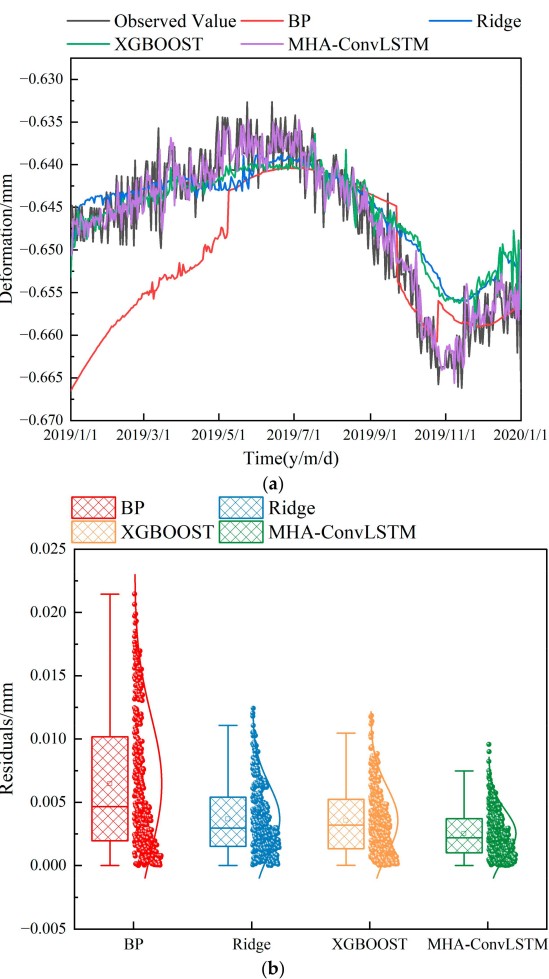

**Figure 14.** (**a**) Forecast of different forecasting models. (**b**) Distribution of residuals for each model prediction.

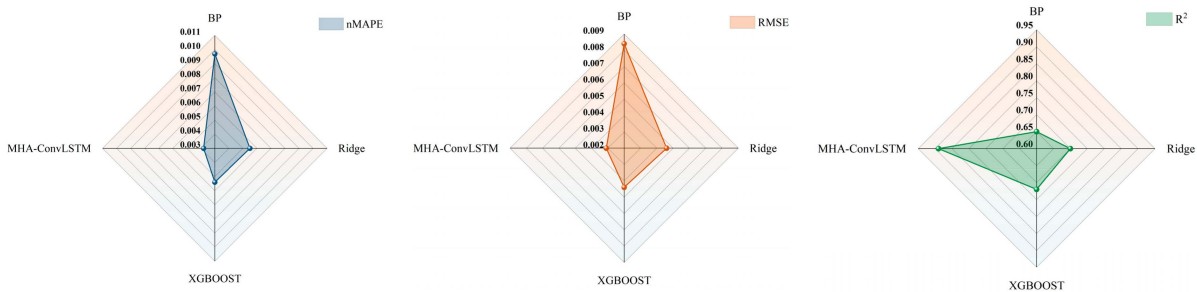

**Figure 15.** Comparison of evaluation indexes of each prediction model.

The combined results demonstrate that, in terms of performance for fitting the nonlinear dam deformation data, the combined MHA-LSTM model greatly surpasses the other widely used models. Among them, BP and Ridge showed model distortion in the early stages of the prediction process for EX2–3 and EX2–4, and the predicted values significantly deviated from the measured values. However, in the middle and later stages of the prediction, the predicted values gradually fit with the measured values, and this situation was tentatively considered to be a poor acceptance of the base model for the information feedback from the attention mechanism. In contrast, the current model may exhibit great fitting effects for each peak, further demonstrating that it can successfully concentrate on the hidden state of the deformation data in the time dimension and then identify their true properties. The prediction performance is stable and excellent in EX2–3 and EX2–4, which

verifies the effectiveness and generalization of the model. Table 4 displays the improvement degree of each model compared with BP. The nMAPE and RMSE are more than 50% better than the plain LSTM model in this set of tests, and the prediction performance is stable and excellent in EX2–3 and EX2–4.

**Table 4.** Lifting degree of each model compared with BP.

| Elevation | nMAPE | RMSE | $R^2$ |
| --- | --- | --- | --- |
| Ridge | 43.3% | 45.2% | 2.2% |
| XGBoost | 45.5% | 47.6% | 4.5% |
| MHA-ConvLSTM | 60.8% | 63.1% | 35.8% |

## 4. Conclusions

This paper presents a hybrid prediction model, MHA-ConvLSTM, for dam deformation that takes into account the lag between environmental factors and deformation. The model builds upon the traditional LSTM for time series prediction by incorporating a multi-headed attention mechanism and convolution technique. Through simulation comparison tests using an active dam as an example, we demonstrated the feasibility of incorporating the lag factor and validated the generalization and effectiveness of the MHA-ConvLSTM model. The main conclusions are as follows:

(1) A hysteresis quantification algorithm is proposed to introduce a new hysteresis factor based on the influence factor of the traditional HST model, aiming to simulate the hysteresis effect of water level on deformation. The experimental results demonstrate that the model with the introduction of the hysteresis factor shows a significant improvement in the prediction accuracy compared with the traditional model;

(2) Compared with the plain LSTM, the MHA-ConvLSTM model is more sensitive to the hidden features of long-time domain deformation data, more sensitive to the temporal features, and more robust compared with other models. This is reflected in the fact that the predicted values fit the measured values more closely and maintain stable performance in the prediction of multiple sets of experimental data, and the evaluation indexes such as RMSE can be improved by more than 50% compared with the traditional model;

(3) The experimental findings demonstrate the applicability of the MHA-ConvLSTM model put forward in this paper for real-world engineering. The hysteresis effect of water level and temperature on deformation can simultaneously be considered in later study, as can hyperparameter optimization.

**Author Contributions:** Conceptualization, H.L.; methodology, H.L.; software, H.L.; validation, H.L.; formal analysis, D.L.; investigation, H.L.; writing—original draft, H.L.; writing—review and editing, D.L.; supervision, Y.D. All authors have read and agreed to the published version of the manuscript.

**Funding:** This research was funded by the National Key Research and Development Program (2022YFC3005502), the National Natural Science Foundation of China (51979174), the National Natural Science Foundation of China Joint Fund (U2040221), and the Special Funds for Basic Research Operations of Central Public Welfare Research Institutes (Y322008).

**Institutional Review Board Statement:** Not applicable.

**Informed Consent Statement:** Not applicable.

**Data Availability Statement:** Not applicable.

**Conflicts of Interest:** The authors declare no conflict of interest.

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
