# Peer review of "MHA-ConvLSTM Dam Deformation Prediction Model Considering Environmental Volume Lag Effect"

_applsci, doi:10.3390/app13148538_

Round 1
Reviewer 1 Report
The paper needs a minor revision. In particular, English and References are weak. For the language, we suggest a review made by a native English speaker. For References, I invite the authors to cite recent results in fractal-wavelet analysis. Thus, I suggest adding the following references (or other ones of the same scientific weight, in accordance with the current MDPI policy).
1. Introducing the Discrete Path Transform (DPT) and its applications in signal analysis, artefact removal, and spoken word recognition. Digital Signal Processing,117(1), 103158, 2021.
2. Fractional-Wavelet Analysis of Positive definite Distributions and Wavelets on D'(C), in Engineering Mathematics II, Silvestrov, Rancic (Eds.), Springer, pp. 337-353,2016.
3. Hyperspectral image classification using wavelet transform-based smooth ordering, Int. J. Wavelets Multiresolut. Inf. Process, 17(6), Article Number: 1950050,2019.
4. Harmonic Sierpinski Gasket and Applications, Entropy, 20(9), 714, 2018.
5. A Framework of Adaptive Multiscale Wavelet Decomposition for Signals on Undirected Graphs, IEEE Transactions on Signal Processing, Volume: 67, Issue: 7,Pages: 1696-1711, 2019.
6. Chebyshev wavelet analysis, Journal of Function Spaces, 2022(1), 5542054, 2022
7. On the Weierstrass-Mandelbrot fractal function, Proc. R. Soc. Lond., Ser. A, vol. 370, no. 1743, pp. 459-484, 1980.
Minor editing of English language required.
Author Response
We are very grateful to you for reviewing the paper so carefully.
In response to your suggestions, we have revised the English as well as the references throughout the text. We have also added the references you suggested.
Reviewer 2 Report
Grammar mistakes and typo errors are present. Please thoroughly review the whole paper.
Author Response
我们非常感谢您如此仔细地审阅论文。
我们已经通读了整篇文章,并根据您的建议纠正了文章中存在的语法错误和拼写错误。
Reviewer 3 Report
This research focus on application of MHA-ConvLSTM models to predict the dam deformation considering environmental volume lag effect. Authors have implemented two models in combined form and analysed that these model perform 50% better than traditional models. A through literature review related to this topic if manuscript is available. This research work is best suited for the publication in the Applied Sciences journal of MDPI. Authors have prepared & formatted manuscript as per the guidelines of the journal. They have presented the paper in order and arrived at proper conclusions. The following comments point to be incorporated before the article is accepted for publication.
· The abstract is not incorporated any numerical values and hence required to add what are the input parameters minimum and maximum values and further to increase results in numerical values of results
· Detailed discussion of the data collection and conversion used need to be presented (the frequency of data collected is one time per day on what basis). A table may be presented about various list of data collected and compared with earlier model results)
· Training data and test data details to be discussed
· In line number 193 check the LSTM.
· Why authors selected these two (MHA-ConvLSTM) alone for deformation prediction.
· A nomenclature is required in order to read easily the article
Author Response
We are very grateful to you for reviewing the paper so carefully.
We have enriched the numerical description of the results in the summary section according to your suggestion 1.
Suggestion 2: The collection frequency of the monitoring instrument is limited by the relevant specifications, and the collection frequency of the selected measurement points in this paper is once a day according to the requirements of the relevant specifications.
Suggestion 3: We have supplemented the detailed description of the division rules of test set and training set in line 313 of the paper.
Suggestion 4: We have checked the LSTM in line 193 and confirmed that the LSTM expression there is correct.
Suggestion 5: Due to the characteristics of dam safety monitoring data such as long time domain and many redundant features, traditional prediction models such as LSTM present problems such as difficulty in memorizing feature selection and poor detail portrayal when facing such data. Therefore, to address the above phenomenon, we choose the Multihead Attention Mechanism model to solve the problem of feature selection and memory, and ConvLSTM to solve the problem of feature learning for time series data.